# A model framework for projecting the prevalence and impact of Long-COVID in the UK

Chris Martin[1]*, Michiel Luteijn[2], William Letton[3], Josephine Robertson[4], Stuart McDonald[5]

**1** UCL Institute of Health Informatics, London, United Kingdom, **2** Hannover Re UK Life branch, London, United Kingdom, **3** Crystallise, Basildon, United Kingdom, **4** Optum International, **5** Lloyds Banking Group, London, United Kingdom

* chris.martin@crystallise.com

**Data Availability Statement:** The model is in an Excel spreadsheet and can be found on the GitHub repository here: https://github.com/Crystallize/longCOVID.

## Abstract

The objective of this paper is to model lost Quality Adjusted Life Years (QALYs) from symptoms arising from COVID-19 disease in the UK population, including symptoms of 'long-COVID'. The scope includes QALYs lost to symptoms, but not deaths, due to acute COVID-19 and long-COVID. The prevalence of symptomatic COVID-19, encompassing acute symptoms and long-COVID symptoms, was modelled using a decay function. Permanent injury as a result of COVID-19 infection, was modelled as a fixed prevalence. Both parts were combined to calculate QALY loss due to COVID-19 symptoms. Assuming a 60% final attack rate for SARS-CoV-2 infection in the population, we modelled 299,730 QALYs lost within 1 year of infection (90% due to symptomatic COVID-19 and 10% permanent injury) and 557,764 QALYs lost within 10 years of infection (49% due to symptomatic COVID-19 and 51% due to permanent injury). The UK Government willingness-to-pay to avoid these QALY losses would be £17.9 billion and £32.2 billion, respectively. Additionally, 90,143 people were subject to permanent injury from COVID-19 (0.14% of the population). Given the ongoing development in information in this area, we present a model framework for calculating the health economic impacts of symptoms following SARS-CoV-2 infection. This model framework can aid in quantifying the adverse health impact of COVID-19, long-COVID and permanent injury following COVID-19 in society and assist the proactive management of risk posed to health. Further research is needed using standardised measures of patient reported outcomes relevant to long-COVID and applied at a population level.

## Introduction

In December 2019, a series of pneumonia cases, now known to be caused by the novel SARS-Cov-2 virus, emerged in Wuhan, China [1]. The novel SARS-Cov-2 virus quickly spread across the globe and on March 11th, 2020, the WHO made the assessment that COVID-19 can be characterised as a pandemic. As of April 2021, the global confirmed death toll stands at over

**Funding:** This study was supported by Crystallise Ltd. in the form of salaries for CM and WL, Lloyds Banking Group in the form of a salary for SM, Hannover Re in the form of a salary for ML, and Optum International in the form of a salary for JR. The specific roles of these authors are articulated in the 'author contributions' section. The funders had no role in study design, data collection and analysis, decision to publish, or preparation of the manuscript.

**Competing interests:** The authors have read the journal's policy and have the following competing interests: SM and WL are employees of Crystallise Ltd, which provides consultancy services to the healthcare, insurance and pharmaceuticals industries. SM is an employee of Lloyds Banking Group, which provides banking, investments, insurance and pension products. ML is an employee of Hannover Re, which reinsures insurance and pension products. JR is an employee of Optum International, which provides healthcare and health insurance services. This does not alter our adherence to PLOS ONE policies on sharing data and materials. There are no patents, products in development or marketed products associated with this research to declare.

1.4 million, with over 150,000 deaths mentioning COVID-19 on the death certificate in the United Kingdom (UK) [2,3].

Over the course of the COVID-19 pandemic, it has emerged that some COVID-19 patients suffer symptoms long after initial infection. The National Institute for Health and Care Excellence (NICE) has defined three phases to symptoms following COVID-19 [4]. First, 'Acute COVID-19 infection' covers the period of active infection up to 4-weeks post-infection. Second, 'Ongoing symptomatic COVID-19' covers the period when infection should have ceased but persisting effects from the infection that may take time to heal may be present from 4 and 12-weeks post-infection. Third, 'Post-COVID-19 syndrome' is defined as '*Signs and symptoms that develop during or following an infection consistent with COVID-19, continue for more than 12 weeks and are not explained by an alternative diagnosis.*' Long-COVID describes both ongoing symptomatic COVID-19 (the second group) as well as post-COVID-19 syndrome (the third group). Documented symptoms for long-COVID include breathlessness, fatigue, myalgia, chest pains and insomnia [5].

Post-COVID-19 syndrome may persist long after active infection has ceased and in some cases symptoms will be permanent. Lung scarring following coronavirus related Acute Respiratory Distress Syndrome (ARDS) or from the high-pressure mechanical ventilation used in its treatment has been widely documented [6]. In a study of patients with acute respiratory distress syndrome about a third of those who were previously employed were still unemployed 5-years later [7], suggesting long term disability. A dysfunctional and uncontrolled immune response can cause multi-organ damage, particularly the liver and kidneys, and disrupt the coagulation control mechanisms of the blood [8]. This can precipitate major adverse cardiovascular events which may have long-term consequences such as heart failure or hemiplegia. Data from the COVID Infection Survey study on long-COVID suggests that the risk of major adverse cardiovascular events is about ten times higher in cases with non-intensive care hospitalized patients with COVID when compared to matched controls [9]. Following treatment in critical care with acute respiratory distress syndrome, about 25% of patients have post-traumatic stress disorder (PTSD) and about 40% suffer depression [10,11]. Severe illness often results in prolonged periods of immobility which range from simple lack of exercise to prolonged bed rest, resulting in further knock-on health impacts. Severe acute respiratory syndrome (SARS) and Middle East respiratory syndrome (MERS) are examples of two other coronavirus outbreaks that have caused similar symptoms to SARS-Cov-2 in the acute stage of infections (i.e., viral pneumonia and ARDS). A recent systematic review and meta-analysis that evaluated the long-term clinical outcomes after SARS and MERS suggest similar symptoms were found 6 to 12-months post discharge, namely reduced lung function, reduced ability to exercise, PTSD, depression, anxiety and reduced Quality of Life (QoL) scores [12].

The objective of this paper is to model lost Quality Adjusted Life Years (QALYs) from both acute COVID-19 and long-COVID symptoms arising from COVID-19 in the UK population. The scope does not include COVID-19 deaths. The parameterisation of the model was based on a literature review. This modelling framework divides the symptomatic cohort into two groups: symptomatic (short-term) COVID and COVID (permanently) injured. The symptomatic COVID group includes all three NICE defined categories described earlier. The assumption is that there are a variety of patterns of illness in the survivors with varying duration and differing aetiologies, but that all are self-limiting and will eventually recover. The COVID-injured group includes people in the post-COVID-19 syndrome group that may have persisting symptoms as a result of permanent injury following infection and associated treatment. These symptoms are assumed to be permanent for the purpose of modelling. The model is presented as a framework, which can be developed as better data for the estimation of parameters

become available, for example for the prevalence of long-term symptoms, the QoL impact of symptoms, and the impact of vaccination programmes.

## Methods

### Model overview

The model developed is a compartmental forecast model, and estimates QALYs lost due to COVID-19 illness, but not deaths, in the UK population of 66.6 million. The baseline model assumed a 60% attack rate at day 0 (39,960,000 persons infected), and no reinfections. Estimates were then made of the proportion of those persons that would be non-surviving. QALYs lost in the surviving persons were then estimated based on the prevalence of symptoms over time. Total symptom prevalence at each time point was estimated as a combination of symptomatic persons with short-term injury, which decays over time, and persons with permanent injury, which has an unchanging prevalence. Short-term symptom prevalence was modelled using a decay function based on the findings of national Coronavirus Infection Survey [9]. Both the short-term injured and the permanently injured were divided into three mutually exclusive treatment groups: non-hospitalised, ward-based care and Intensive Treatment Unit (ITU) care (Fig 1). COVID-19-related QALY loss differed by treatment group, while the probabilities of emerging with permanent injury also varied by treatment group.

Once the estimates had been made for the proportions of the subjects that follow each treatment/severity stream, and the QALYs lost by those subjects for each treatment/severity stream, the core of the model is was a calculation of the cumulative days lived with symptoms and/or permanent injury up to the modelled time-horizon, multiplied by the number of QALYs lost per day for those with symptoms, and discounted over time. The time horizon was set at the life expectancy for both the Symptomatic COVID and COVID-injured cohorts. Taking account of the age distribution of people admitted with COVID-19 [13], the population weighted average life expectancy for them as of 2019 was 19.19 years (own calculations). It is expected that those admitted are in poorer health than the population average and so a reduction factor of 50% was applied to reach a time horizon of 10 years for hospitalised patients. This is in-line with Briggs et al, who estimate the life expectancy of the average UK COVID-19 death at 10.94 years [14].

The model has been made public and can be found on the Github repository at this location: https://github.com/Crystallize/longCOVID.

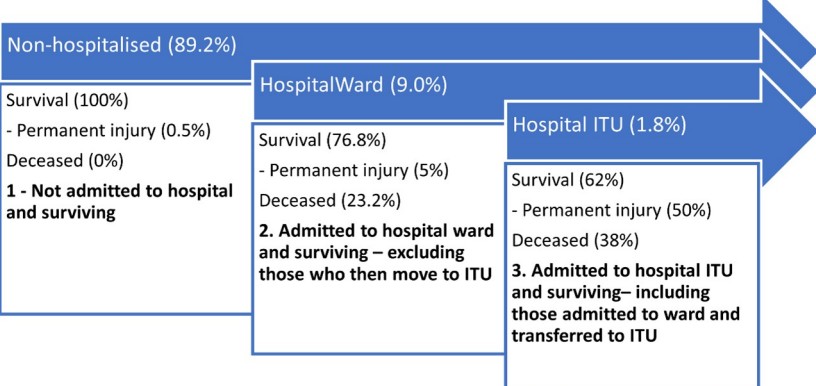

**Fig 1. The pathways of care for the three survivor compartments amongst symptomatic patients.**

## Model parameters

**Prevalence of symptoms.** UK surveys of prevalence of symptoms show a range of outcomes, with hospital-based surveys [5,15] showing higher and an app-based survey [16] showing a lower prevalence of symptoms than the national Coronavirus Infection Survey [9] (Fig 2). The national Coronavirus Infection Survey was conducted by the Office of National Statistics (ONS). This survey was carried out between April and December 2020 and included 8,193 respondents. It consisted of a random, a-priori, selected sample that were invited for COVID tests and therefore would cover asymptomatic and symptomatic cases. The results of this survey were considered most relevant as they contains the largest sample and provides a reference to the UK general population prevalence. Two key data points provide the prevalence of symptoms at 5 and 12-weeks, which were approximately 20% and 10% respectively. A natural history of symptom prevalence over time was fitted to all infections in the model by fitting an exponential decay curve on these two data points, plus a 50% symptom prevalence assumed at t = 0 (Eq 1).

$$P_{SymptomaticCOVID,t} = 0.4548 * e^{-0.132*t} \tag{1}$$

Where:

- $P_t$ is the prevalence of any symptoms at time t in weeks following infection.

- Decay function constant = 0.4548 and rate of decay = -0.132 derived using Coronavirus Infection Survey study data points.

**Distribution of groups and group mortality outcomes.** The number of UK positive tests (2,657,305) and hospital admissions (287,662) with COVID-19 up to 31st October 2020 indicated 10.8% of known cases were admitted to hospital [3]. We assumed a similar distribution for the modelled symptomatic cases. Assuming the mortality rate in the non-hospitalised group was negligible, the surviving non-hospitalised fraction of all positive tests was 89.2%. Of those admitted to hospital (10.8%), 16.5% of these were admitted to critical care [17] leaving 9.0% of all cases who were admitted for ward-care only. The mortality rate for critical care

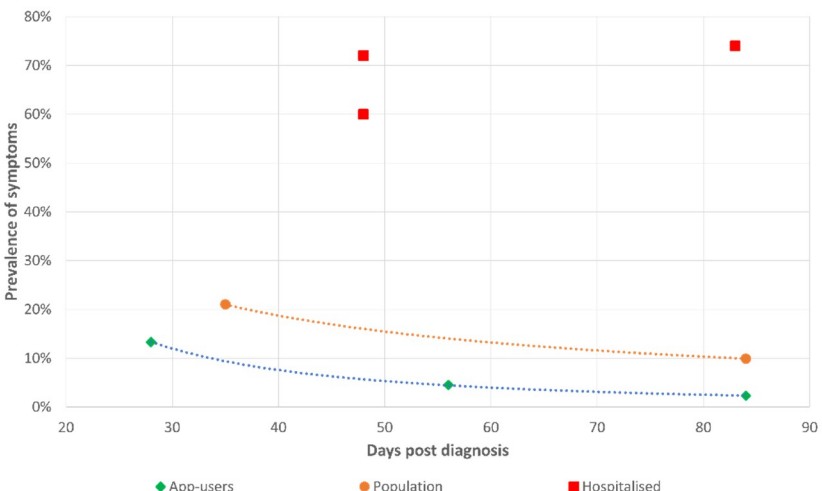

**Fig 2. Symptom prevalence across studies identified in the UK by duration and severity group.**

patients was 38% in October 2020 [18] meaning 62% of the 1.8% of cases admitted to critical care survived (1.1%). As of 31st August 2020 there had been 118,613 patients admitted to hospital in England, Wales and Northern Ireland (excluding ITU). Of those, 27,483 died on the wards making the ward mortality 23.2% with 76.8% surviving [3,19]. Therefore, 6.9% of the 9.0% admitted for ward-care survived.

**Prevalence of permanently injured.** The proportion of known COVID-19 cases that are permanently injured is not yet known. It is assumed that only patients with positive COVID-19 tests that had symptoms at 6 weeks post-COVID infection can get permanent injury. Based on an estimated 12% of the 66.6 million UK population having been infected [20] and 2.66 million positive COVID-19 tests [3] as of December 31st 2020, 33% of SARS-Cov-2 infections resulted in a positive test (assuming no reinfections). Symptom prevalence at 6 weeks was set at 72% for ITU and 60% for ward patients [15], while for non-hospitalised patients it was set at 16% (interpolated from the COVID Infection survey [9]).

About a third of previously employed patients with ARDS were still unemployed 5-years later [7]. Here it was assumed that disability arising from the illness was the sole cause of unemployment and that a fraction who are injured may manage to return to work, implying an injured figure higher than 33%. Reflecting this uplift, here it was estimated that 50% of the ITU survivors with symptoms at 6 weeks post-COVID will be left permanently injured. For the ward-based group and the non-hospitalised group, here it was assumed 5% permanently injured and 0.5% permanently injured amongst survivors symptomatic at 6 weeks post-COVID, respectively (10% and 1% of the ITU rate, respectively). Using the breakdown between non-hospitalised (89.2%), ward (9%) and ITU (1.8%) populations for symptomatic cases, taking into account deaths in the latter two groups, resulted in a weighted prevalence of 0.62% permanently injured amongst COVID-19 cases with positive tests. Adjusting for positive tests only resulted in a final 0.2% prevalence of the permanently injured amongst all COVID-19 cases.

**Utility.** Loss of utility due to COVID-19 was defined as the change in EQ5D score, which is a questionnaire-based measure of five well-being dimensions: mobility, self-care, usual activities, pain/discomfort, and anxiety/depression. Average loss of utility for hospitalised COVID-19 patients, split by general ward-care (-6.1%), and care on ITU treatment (-15.5%) was reported in the UK after a mean of 48 days [15]. For the purpose of the model, the prevalence of persisting symptoms was taken as that of the prevalence of persistent fatigue (72% for ITU patients and 60% for ward-only patients) on the assumption that the vast majority of other symptoms co-exist with fatigue. The reported average utility change was converted into the COVID-19 symptomatic utility change using the persisting symptom prevalence. This resulted in a COVID-19 symptomatic utility change of -6.1%/60% = -10% for ward patients and -15.5%/72% = -22% for ITU patients symptomatic at 48 days post COVID-19. It was not possible to source utility for a UK non-hospitalised population. We assumed the utility loss for non-hospitalised COVID-19 patients with persistent symptoms to be the same as for the ward-based patients at -10%.

The utility for the permanently injured group was taken from a secondary health-economic analysis of a randomised controlled trial of 795 ARDS patients ventilated in critical care in the UK [21]. At one-year post-discharge, mean utility was 0.58, both for those above and below the age of 65. Taking into account the reference population utility for the UK (0.856) [22], the ARDS specific utility at 1-year was calculated at 0.58/0.856 = 0.68.

Aggregate COVID-specific utility for the symptomatic COVID cohort was calculated at -11%, using a weighted sum of the utilities for the three treatment groups (ward, ITU and population).

### Sensitivity analysis

Univariate sensitivity analysis was carried out in order to assess the effect of the estimates and assumptions of key model parameters on the model output of QALYs lost. Where appropriate, the baseline parameter value was perturbed by 20% in either direction. The first exception was the time horizon parameter, where values of 1 year and 20 years were used either side the baseline value of 10 years. The second exception was the percentage prevalence of symptoms, where the values for each of ITU, Ward, and Outpatient were first converted to odds before the 20% perturbation and then converted back to percentage.

Model parameters, their sources, and the values used for the sensitivity analysis are summarised in Table 1.

## Results

We modelled QALY loss due to COVID-19 symptoms, but not deaths. Following infection, QALY-loss due to symptomatic COVID-19 increases with time, but quickly levels off as people recover. However, for those living with permanent damage, QALY-loss accumulates over their life expectancy. Within a 1-year time-horizon, the estimated undiscounted QALY loss in survivors was 299,730 (0.6% of the total expected QALYs for that year) with 271,037 QALYs (92%) lost to symptomatic COVID-19 in the acute, ongoing and post-COVID syndrome; and 28,692 (8%) lost to permanent injury from COVID-19. Discounted QALY loss was 298,942, representing a monetary value of £17.9 billion based on the UK Government's willingness-to-pay per QALY [23], and an average loss of about 0.0075 QALY per infection. With a 10-year time-horizon, the estimated total undiscounted QALY loss in survivors was 557,764 with 271,310 (54%) QALYs lost to symptomatic COVID-19 in the acute, ongoing and post-COVID syndrome, and 286,454 (46%) lost to permanent injury from COVID-19. Discounted QALY loss was 536,877, representing a monetary value of £32.2 billion and an average loss of about 0.013 QALY per infection. Regardless of timeframe, an estimated 90,142 people would be left with permanent injury. Estimates of QALYs lost up to 10 years, with or without the 1.5% annual discount rate, are shown in Table 2, and represented graphically in Fig 3.

Sensitivity analyses were performed on parameter values in order to assess the robustness of the model over the illustrative 10-year time horizon (Fig 4). Unsurprisingly, discounted QALY loss is sensitive to the time horizon considered. However, as the majority of the QALY loss occurs in the first year, reducing the time horizon from 10 to 1-year reduced discounted QALY loss by 44.3%. A reduction or increase in attack rate directly translates into a similar reduction or increase in QALY loss respectively. The model is less sensitive to parameters to do with QALY loss and prevalence for symptomatic COVID-19 and permanent injury as QALY loss is split between these two conditions. On shorter timeframes, the model would be more sensitive to assumptions around symptomatic COVID-19 prevalence and QOL loss as symptomatic COVID-19 is a larger proportion of total QALY loss at shorter timeframes.

## Discussion

We modelled QALY loss due to COVID-19 symptoms and permanent injury in the UK population. To the best of our knowledge, this is the first such study on a UK population using UK data. Basu and Gandhay modelled the QALY impact of averting a single COVID-19 infection in an American setting [24] and reported QALY loss due to symptomatic (outpatient) COVID-19 of 0.007 (95% CI: 0.002–0.011) per COVID-19 infection. This compares to our 0.0075 and 0.0135 for 1- and 10-year horizons, respectively. A further 0.002 QALY loss to family members due to symptomatic COVID-19 and 0.048 QALY due to COVID-19 deaths was modelled by Basu and Gandhay, both of which was out of our scope. Basu and Gandhay

**Table 1. Key parameter values: Baseline and sensitivity tested.**

| Parameter | Baseline value | Source | Evidence strength | Sensitivities tested |
|---|---|---|---|---|
| **Infection attack rate** | 60% | Results of an age-stratified, susceptible, exposed, infected, recovered and died (SEIRD) model (own calculations). | key assumption update as risk of infection varies. Availability of testing may impact figures. | 48–72% |
| **Prevalence function for prevalence of symptoms post COVID-19 by the day.** | $P_S = C.e^{-\lambda.\text{days}}$ C = 0.4548 l = 0.132 | Fitted to the results of the Coronavirus Infection Survey long-COVID report December 2020 [9] plus an assumed 50% symptoms at t = 0. | key assumption update as evidence emerges | Constant term 0.3638–0.5458 |
| **Prevalence of symptoms at 6-weeks for survivors of ITU.** | 72% | [15] | | |
| **Prevalence of symptoms at 6-weeks in ward-care only survivors.** | 60% | [15] | | |
| **Proportion of ITU survivors with persistent symptoms at 6-weeks who are permanently injured.** | 50% | An assumption based on the observation that 33% of those employed at the time of admission to ITU with ARDS are still unemployed 5-years later [7] with an up-lift applied to reflect those returning to work while permanently injured. | Placeholder estimate to be updated when evidence emerges | 40–60% |
| **Proportion of ward-care survivors with persistent symptoms at 6-weeks who are permanently injured.** | 5% | An assumption that the prevalence is 10% of the ITU prevalence. | | |
| **Proportion of non-hospitalised cases with persistent symptoms at 6-weeks who are permanently injured.** | 0.5% | An assumption that the prevalence is 10% of the ward-care prevalence. | | |
| **Proportion of all known cases surviving critical care.** | 1.1% | Calculated from the proportion of cases admitted to ITU and the survival rate on ITU. | Survival and hospitalisation rates may change as treatment improves, vaccine reduce disease risk, virus variants impact fatality. | |
| **Proportion of all known cases surviving ward.** | 6.9% | Calculated from the proportion of cases admitted to a hospital ward and the survival rate on the ward. | | |
| **Proportion of all known cases that are non-hospitalised that survive.** | 89.2% | The proportion of cases not admitted on the assumption that the mortality rate is negligible in this group. | | |
| **Adjusted prevalence of permanent injury for all infections, known and unknown.** | 0.226% | Calculated from the prevalence of permanent injury in known cases and the proportion of all infections that are identified as cases. | key assumption update as evidence emerges | 0.182–0.273 |
| **Utility loss for all symptomatic cases** | 0.103 | Derived from weighting the average utility loss for symptomatic ward and ITU survivors at 6 weeks [15]. Symptomatic non-hospitalised patients are assumed to have similar utility loss as symptomatic ward patients. | | 0.082–0.123 |
| **Utility loss for those left with permanent injury post-COVID.** | 0.318 | Calculated from the utility loss at 1-year post ITU discharge for ARDS [21] and the population norm for England [22]. | Evidence will need to be accumulated for COVID-19 | 0.254–0.381 |
| **Time horizon (years)** | 10 | Assumption based on adjusted weighted population life expectancy for COVID-19 hospital admissions. | Key assumption update as evidence emerges. | 1–20 |
| **Annual discount rate for future QALYs** | 1.5% | [23] | | |
| **Monetary value per QALY** | £60,000 | [23] | | |

assumed utility loss for symptomatic outpatients of 0.43 (based on utility of H1N1 patients on the day of index medical visit), compared to our 0.10, based on COVID-19 symptomatic patients on average 48 days post-discharge. Given the longer timeframe of our model (including modelled symptomatic infections), we felt 0.10 is appropriate. In Basu and Gandhay's

**Table 2. Estimates of total QALYs lost over 1 and 10 year time horizons.**

| Time horizon (years) | QALYs lost | | | |
|---|---|---|---|---|
| | **Permanent injury** | **short-term injury** | **Total** | **Total (discounted)** |
| 1 | 28,692 | 271,037 | 299,730 | 298,942 |
| 2 | 57,385 | 271,310 | 328,695 | 327,269 |
| 3 | 85,999 | 271,310 | 357,309 | 354,837 |
| 4 | 114,613 | 271,310 | 385,923 | 381,997 |
| 5 | 143,227 | 271,310 | 414,537 | 408,757 |
| 6 | 171,919 | 271,310 | 443,230 | 435,193 |
| 7 | 200,533 | 271,310 | 471,844 | 461,167 |
| 8 | 229,147 | 271,310 | 500,458 | 486,757 |
| 9 | 257,761 | 271,310 | 529,072 | 511,970 |
| 10 | 286,454 | 271,310 | 557,764 | 536,877 |

model, 0.005% of symptomatic patients recover with permanent kidney injury, while in our model, 0.2% of all infections resulted in permanent injury, with a wider consideration of injury.

Various studies have reported widely varying estimates of symptom prevalence (Fig 2). At 12 weeks, the Coronavirus Infection Survey (shown as Population) reported symptom prevalence of 9.9%, while the Arnold study reported a prevalence of 74% (shown as Hospitalised). The Arnold study sample is restricted to hospitalised patients, whereas the Coronavirus Infection Survey study is population-based. Even when considering hospitalisation as a risk factor and the different study populations, the prevalence variation is striking. The Sudre study (shown as App-users) conducted from a COVID app reported a lower symptom prevalence of 2.3%. This might reflect sampling and recording biases as the users were self-selected and responsible for recording symptoms. The means of eliciting responses in symptom studies can significantly impact estimated prevalence thus making comparison between studies difficult [25–27].

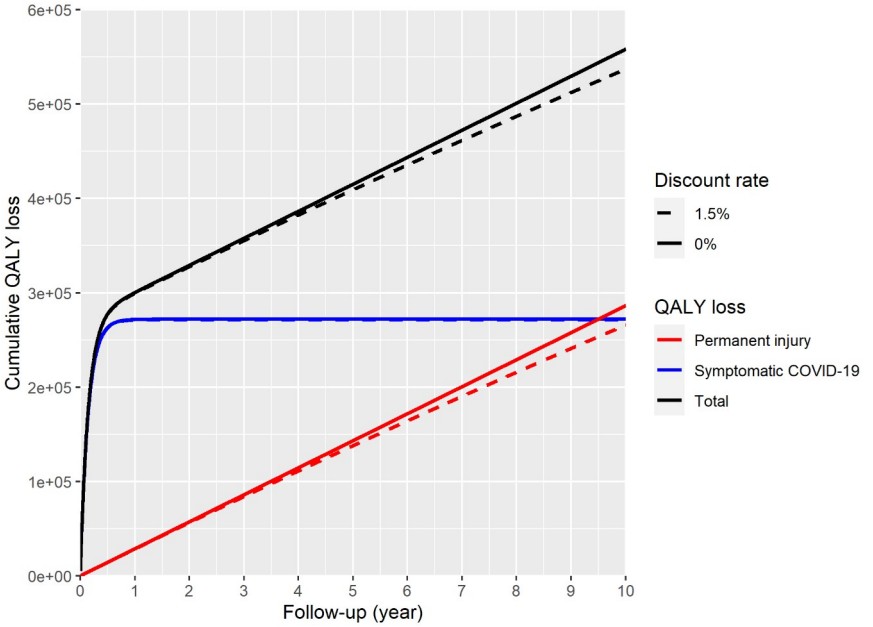

**Fig 3. Cumulative QALY loss for symptomatic COVID and permanent injury due to COVID.**

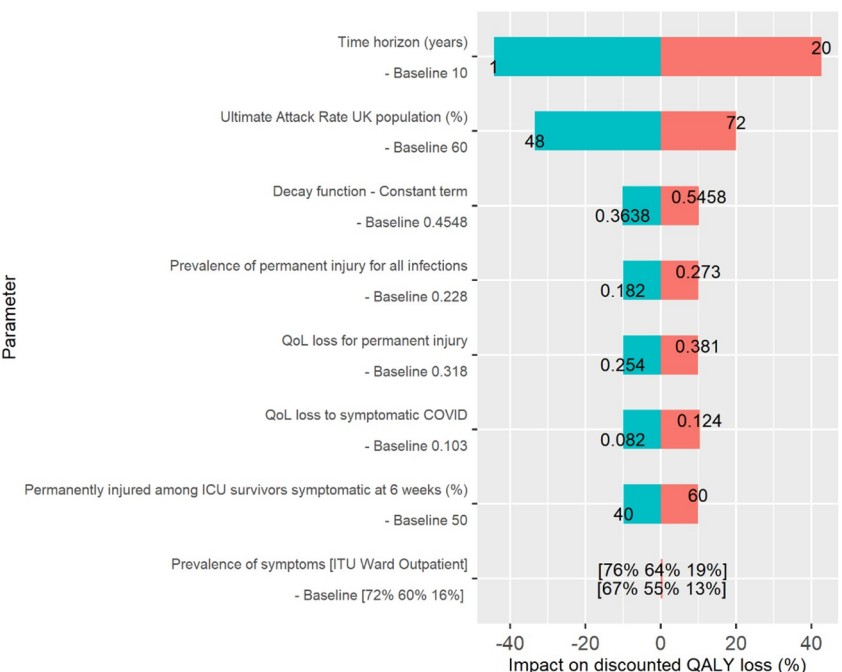

**Fig 4. Results of sensitivity analysis on key parameters.**

Symptom prevalence studies are also complicated by adjusting for background prevalence as well as varying definitions of symptoms. We used the Coronavirus Infection Survey study to inform the prevalence of ongoing symptoms from diagnosis since it was the study with the largest sample and reflected the general population prevalence due to being population-based.

To improve understanding of long-COVID going forward, the quality of the symptom prevalence data could be improved through the use of a standardised measurement and recording of symptoms across studies. Currently, data is being gathered using different types of questionnaires in different mediums, e.g. Halpin (2020) [15] developed their own COVID-19 rehabilitation telephone screening tool, Arnold (2020) [5] used the SF-36 questionnaire, and Sudre (2020) [16] used a self-reporting questionnaire via an app. Standardised and validated questionnaires and tools such as St George's respiratory questionnaire and the MRC dyspnoea scale [28,29] are used to record patient reported outcomes. However, these tools are often disease-specific and may not be appropriate for use in long-COVID patients. In addition, HRQoL questionnaires like SF-36 maybe too general and may not capture all effects resulting from the multiple possible symptoms of COVID-19. Finally, without pre-COVID-19 baseline measurements from the subjects, symptom data may be subject to recall bias.

The effects of COVID-19 are not limited to health and the economic impacts to individuals and the country, as well as the wellbeing impacts on family members of those symptomatic were outside of the scope of this study. Concurrent to disease impact, the population are living with non-pharmaceutical interventions which limit the spread of SARS-Cov-2 but are also known to impact wellbeing directly and indirectly [30].

## Potential implications

**Health and care services.** Proactive care and tailored intervention support will be required in order to locate and accommodate the needs of the COVID-injured in the most appropriate setting. We second the recommendation from Halpin et al. that rehabilitation

services should be planned "to manage these symptoms appropriately and maximise the functional return of COVID-19 survivors." [15] There will be a lasting health burden within our society for those who are COVID-injured who will require ongoing support.

Prevention is better than cure. We provide these numbers as health economic rationale or a willingness-to-pay to avoid an accumulation of injury due to COVID-19. This provides further justification for the vaccination programme, which has been shown to provide significant reduction in severe disease outcomes [31,32]. Given the socio-economic disparity in the pandemic burden, this may provide justification for spending which seeks to reduce health inequalities such as tailored public health messaging, vaccination delivery, and community access to care. In addition, this provides support for non-pharmaceutical interventions that reduce the transmission of the SARS-Cov-2, such as physical distancing and the use of face masks.

**Societal and economic.**   For the Symptomatic COVID-19 cohort, return to work may be delayed causing increased claims on statutory sick pay, group employer or individual income protection insurance. For the COVID-injured cohort, some may not return fully to work. This may increase claims on government unemployment and disability benefits. Both cohorts would benefit from flexibility in working arrangements and return to work to better accommodate the individuals' needs and ensure continued employability.

Given the socio-economic disparity in the pandemic burden, there may be disparity in the economic impact of permanent injury from COVID-19 which warrants further investigation. Some in society have been, and are, at increased risk of infection due to their occupation. We agree with the calls for "further research into the role of repeated exposure to SARS-Cov-2 in a healthcare delivery setting and or in the community, and role of the repeated exposures leading to autoimmune mediated responses" [33].

## Limitations

The mechanisms of underlying pathogenesis and resulting symptoms of COVID-19 is not yet fully understood. Although NICE has published a working definition, this may be subject to change. This model estimates a disease that is evolving and as such, its ability to predict long-term outcomes will be limited. There is uncertainly around some of the parameters being used in the model and a number of assumptions had to be made. Survival rates of ward and ITU care were based on 2020 data and could since have changed as improvements are being made in care for COVID-19 patients. Improved knowledge on treatment for COVID-19 in wards and ITUs could also reduce the proportion of permanently injured amongst survivors.

Our calculation of infections that result in long-COVID uses fatigue as the most common symptom post-discharge [15], as it was most commonly reported in symptom prevalence studies. However, fatigue is also a commonly reported symptom in the general population and prevalence varies. One review of fatigue as a symptom in 1992 found prevalence estimates in the general population ranging from 4% to 45% (26). Therefore, reported 'fatigue' could be due to factors other than long-COVID.

One of the assumptions of the model is that the average lost QALY rate for symptomatic patients in shorter durations is similar to the long-COVID lost QALY rate. This will affect how well the lost QALY rate estimated by the model reflects the actual HRQoL of long-COVID. As more data is collected on HRQoL in patients with persistent symptoms, these limitations can be reconciled.

## Conclusion

This article describes a model for estimating the health impact of COVID-19 symptoms, including symptomatic (short-term) COVID and COVID (permanently) injured. Quality

adjusted life-years lost are used to present a standardised measure of the impacts and uses the UK government's willingness-to-pay metric to quantify the impact in monetary terms. The model framework is presented such that it can be updated with information as more reliable data accumulates. Based on the current parameterisation, 557,764 QALYs would be lost over 10-years, 286,454 to permanent injury as a result of COVID-19 and 271,310 from symptoms of COVID-19 across all timescales. This corresponds to an average loss of 0.013 QALY per infection. An estimated 90,142 people could be left living with significant impairments as a result of injury from COVID-19.

This model framework highlights just some of the factors that will influence the impact of the Long-COVID burden in our society, our limited understanding of the condition to date, and the limited information available. There is great uncertainty in the prevalence of symptoms over time as a result of lack of standardisation in methods used to measure it. A standardised patient report outcomes instrument could aid understanding in this area and would require development and validation specific to COVID-19 symptoms.

## Acknowledgments

Many people have supported this work, including:

- Steve Bale, Senior Actuary, Munich Re

- Scott Reid, Global Protection Pricing & Product Development Actuary, Zurich Insurance

- Matt Gurden, Actuarial Director, Government Actuary's Department

- Louisa Rutherford, Medical writer, Crystallise

- Holly Gould, Researcher, Crystallise

- Hannah Rice, Researcher, Crystallise

- Walter Rodney Ngumo, Researcher, Crystallise

- Nicola Clarke, Researcher, Crystallise

- Christian Todaro, Researcher, Crystallise

- Colin Dutkiewicz, Global Head of Life, Aon Reinsurance Solutions

## Author Contributions

**Conceptualization:** Chris Martin, William Letton, Stuart McDonald.

**Formal analysis:** Chris Martin.

**Methodology:** Chris Martin, William Letton.

**Project administration:** Michiel Luteijn.

**Supervision:** Stuart McDonald.

**Validation:** Chris Martin, Michiel Luteijn, William Letton, Josephine Robertson.

**Visualization:** Michiel Luteijn.

**Writing – original draft:** Chris Martin.

**Writing – review & editing:** Chris Martin, Michiel Luteijn, William Letton, Josephine Robertson, Stuart McDonald.

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
