## [Decision Letter · Decision Letter 0]

5 Jul 2021

PONE-D-21-16088

A model framework for projecting the prevalence and impact of Long-COVID in the UK

PLOS ONE

Dear Dr. Martin,

Thank you for submitting your manuscript to PLOS ONE. After careful consideration, we feel that it has merit but does not fully meet PLOS ONE’s publication criteria as it currently stands. Therefore, we invite you to submit a revised version of the manuscript that addresses the points raised during the review process.

Please address the issues and revise accordingly.

We look forward to receiving your revised manuscript.

Kind regards,

Academic Editor

PLOS ONE

Journal Requirements:

2. We note that longCOVID/LICENSE data presented as the supporting information on GritHub in your submission contain copyrighted images/data. All PLOS content is published under the Creative Commons Attribution License (CC BY 4.0), which means that the manuscript, images, and Supporting Information files will be freely available online, and any third party is permitted to access, download, copy, distribute, and use these materials in any way, even commercially, with proper attribution. For more information, see our copyright guidelines: http://journals.plos.org/plosone/s/licenses-and-copyright.

2.1. You may seek permission from the original copyright holder of Figure/Data to publish the content specifically under the CC BY 4.0 license.

2.2.    If you are unable to obtain permission from the original copyright holder to publish these figures under the CC BY 4.0 license or if the copyright holder’s requirements are incompatible with the CC BY 4.0 license, please either i) remove the figure or ii) supply a replacement figure that complies with the CC BY 4.0 license. Please check copyright information on all replacement figures and update the figure caption with source information. If applicable, please specify in the figure caption text when a figure is similar but not identical to the original image and is therefore for illustrative purposes only.

Reviewers' comments:

Reviewer's Responses to Questions

**Comments to the Author**

1. Is the manuscript technically sound, and do the data support the conclusions?

Reviewer #1: Partly

Reviewer #2: Yes

2. Has the statistical analysis been performed appropriately and rigorously? 

Reviewer #1: No

Reviewer #2: I Don't Know

3. Have the authors made all data underlying the findings in their manuscript fully available?

Reviewer #1: Yes

Reviewer #2: Yes

4. Is the manuscript presented in an intelligible fashion and written in standard English?

Reviewer #1: Yes

Reviewer #2: Yes

5. Review Comments to the Author

Reviewer #1: Reviewer’s comments

Method

• The authors should provide the information first about the survey, study population and the study groups clearly.

• Define the symptomatic COVID-19 cases and permanently injured cases

• Define the three subgroups of symptomatic COVID-19 as well.

• According to recent devlepment on COVID-19 symptomatic cases, there are four types : mild, moderate, severe (hospitalized), very severe (in ICU). Why did the authors did not adopt these subgroups of symptomatic cases ?

• Authors dont define what is long COVID ?

• Please add a subsection of « sensitivity analysis » performed in the method section.

Model :

In the method section, some essential information are missing and it is not easy to understand the method section without this information :

• It is unclear which model was used to creat the framework ?

• The rational to choose the decay function and to keep the fixed prevalence for permanently injured group

• Please add the abbrevaitions in the text such as ITU, CIS study, PTSD, etc.

Utility

• Please define « average utility change » ?

Table 1 is describing the parameters used for the model and also it reported the result of the sensitivity anaysis that should be the part of the results section ?

Results

Overall the result section seems to have insufficient information in interpretation, no table is provided.

• It is unclear which framework is developed ?

• The authors can provide a comparison of discounted and undicounted estimates and report them in a table at two time points of 1-year and 10 year time horizen.

• In the method section, authord did not report that the permanent injury from COVID-19 including lung fibrosis, the sequelae of major adverse cardiovascular events like heart attacks and strokes and psychological impacts such as PTSD, will be measured ? This information should be reported in the method section first.

Discussion

This section is very long and less convincing.

Reviewer #2: I found the article quite interesting and well written. However, I will admit that though I am familiar with QALYs and many of the tools used to collect data contributing to QALY calculations, my understanding of the statistics behind these analyses is very weak. Therefore I could not properly assess the statistical analyses of this work.

Regarding the methods section, I noticed the tense was often changed to the present tense. This is very confusing and I feel it is important to remain with the past tense at all times, as the methods are describing what you did. This is true as well for the description of your methods in the abstract. Therefore, "Both parts are combined..." should be changed to "Both parts were combined" on line 28.

This is a very minor comment, but could be of interest. You use UTI, which I assume is Intensive Treatment Unit. As this is often referred to as the ICU in the United States, it may be of interest to simply define it the first time you use this acronym (I believe on line 99).

In the Discussion section, on line 263 you mention, "In addition, this provides support for non-pharmaceutical interventions that reduce the transmission of the SARS-Cov-2", this is a great point. It could be of interest to provide one or several examples by finishing the sentence with, "such as..."

There appears to just be a small typo on line 267 "not return to fully to work" should be "not return fully to work"

This is very small, but on line 287 it should be "different types of questionnaires" with an "s" at the end

On line 279, I believe it should be "although NICE has published" instead of "have published"

The limitations section made some very interesting points. I did feel though this section could be re-organised slightly. For example, on line 307 the discussion of recall bias is very interesting but does not fit there. I would discuss recall bias under paragraph 2 where you discuss challenges with standardized measures and recording of data.

Overall I found the article to be quite interesting and very well written. Bravo!

6. PLOS authors have the option to publish the peer review history of their article (what does this mean?). If published, this will include your full peer review and any attached files.

Reviewer #1: No

Reviewer #2: **Yes: **Caroline Barnes

---

## [Author Response · Author response to Decision Letter 0]

4 Sep 2021

Reviewer #1: 

 Method

* The authors should provide the information first about the survey, study population and the study groups clearly.

We are not certain to what this refers. 

Additional information has been included on the ONS COVID symptom survey and study population:

 "The national Coronavirus Infection Survey was conducted by the Office of National Statistics (ONS). This survey was carried out between April and December 2020 and included 8,193 respondents. It is considered most relevant as it contains the largest sample and provides a reference to the UK general population prevalence. The Coronavirus Infection Survey consisted of a random, a-priori, selected sample that is were invited for COVID tests and therefore would cover asymptomatic and symptomatic cases."

 The methods section has been amended to be clearer as to the population and subpopulations used in our modelling process:

 “Both the short-term injured and the permanently injured were divided into three mutually exclusive treatment groups: non-hospitalised, ward-based care and Intensive Treatment Unit (ITU) care (Fig 1). COVID-19-related QALY loss differed by treatment group, while the probabilities of emerging with permanent injury also varied by treatment group.”

* Define the symptomatic COVID-19 cases and permanently injured cases

This has been clarified:

 “Total symptom prevalence at each time point was estimated as a combination of symptomatic persons with short-term injury, which decays over time, and persons with permanent injury, which has an unchanging prevalence.”

* Define the three subgroups of symptomatic COVID-19 as well.

This section in the introduction has been amended to make it clearer how the definition of long-covid corresponds to the three symptom phases defined by NICE:

 “The National Institute for Health and Care Excellence (NICE) has defined three phases to symptoms following COVID-19 [4]. First, ‘Acute COVID-19 infection’ covers the period of active infection up to 4-weeks post-infection. Second, ‘Ongoing symptomatic COVID-19’ covers the period when infection should have ceased but persisting effects from the infection that may take time to heal may be present from 4 and 12-weeks post-infection. Third, ‘Post-COVID-19 syndrome’ is defined as ‘Signs and symptoms that develop during or following an infection consistent with COVID-19, continue for more than 12 weeks and are not explained by an alternative diagnosis.’ Long-COVID describes both ongoing symptomatic COVID-19 (the second group) as well as post-COVID-19 syndrome (the third group). Documented symptoms for long-COVID include breathlessness, fatigue, myalgia, chest pains and insomnia [5].”

* According to recent devlepment on COVID-19 symptomatic cases, there are four types : mild, moderate, severe (hospitalized), very severe (in ICU). Why did the authors did not adopt these subgroups of symptomatic cases ?

The four types appear to be arbitrarily defined and in any case do not appear to relate directly to the persistence of symptoms but rather the diagnosis and treatment streams. However, our manuscript does distinguish non-hospitalised, ward-based- and ICU patients as distinct groups that are both well-defined and for which some symptom persistence data was available.

* Authors don’t define what is long COVID ?

The introduction now references the definition of long-covid to the three symptom phase phases given by NICE:

 “Long- COVID describes both ongoing symptomatic COVID-19 (the second group) as well as post-COVID-19 syndrome (the third group). Documented symptoms for long- COVID include breathlessness, fatigue, myalgia, chest pains and insomnia [5].”

* Please add a subsection of < sensitivity analysis > performed in the method section.

Added.

Model :

*In the method section, some essential information are missing and it is not easy to understand the method section without this information :

* It is unclear which model was used to create the framework ?

This has hopefully now been made clearer at the start of the methods section:

“The model developed is a compartmental forecast model, and estimates QALYs lost due to COVID-19 illness, but not deaths, in the UK population of 66.6 million.”

The ‘model overview’ section gives a broad structure of the model, and then the ‘model parameters’ section goes into further detail.

Additionally, a sentence has been added to the introduction explaining the use of the term ‘framework’ in the manuscript:

 “The model is presented as a framework, which can be developed as better estimates for parameterisation become available, for example for the prevalence of long-term symptoms, the QoL impact of symptoms, and the impact of vaccination programmes.”

* The rational to choose the decay function and to keep the fixed prevalence for permanently injured group

Again, hopefully this is now clearer:

 “QALYs lost in the surviving persons were then estimated based on the prevalence of symptoms over time. Total symptom prevalence at each time point was estimated as a combination of symptomatic persons with short-term injury, which decays over time, and persons with permanent injury, which has an unchanging prevalence. Short-term symptom prevalence was modelled using a decay function based on the findings of national Coronavirus Infection Survey [9].”

* Please add the abbrevaitions in the text such as ITU, CIS study, PTSD, etc.

Full terms for UK, NICE, ARDS, PTSD, QOL, QALY, ONS, ITU, HRQoL, and PRO have all been added/checked. The reference to CIS study for Coronavirus Infection Survey was removed. 

Utility

* Please define < average utility change > ?

Hopefully this is now clearer:

 “Loss of utility due to COVID-19 was defined as the change in EQ5D score, which is a questionnaire-based measure of five well-being dimensions: mobility, self-care, usual activities, pain/discomfort, and anxiety/depression.”

Table 1 is describing the parameters used for the model and also it reported the result of the sensitivity anaysis that should be the part of the results section ?

This table does not report results of the sensitivity analysis, only the range of values used for the purpose of sensitivity analysis. We have altered the column headings to make this clearer.

Results

*Overall the result section seems to have insufficient information in interpretation, no table is provided.

* It is unclear which framework is developed ?

Hopefully this has been addressed in the changes to the Model section:

 “The model developed is a compartmental forecast model, and estimates QALYs lost due to COVID-19 illness, but not deaths, in the UK population of 66.6 million.”

* The authors can provide a comparison of discounted and undicounted estimates and report them in a table at two time points of 1-year and 10 year time horizen.

Done. Table 2 contains estimates for years 1-10 as a companion to the graphical representation in Figure 3.

* In the method section, author did not report that the permanent injury from COVID-19 including lung fibrosis, the sequelae of major adverse cardiovascular events like heart attacks and strokes and psychological impacts such as PTSD, will be measured ? This information should be reported in the method section first.

The nature of permanent injury due to COVID-19 infection and treatment is discussed in the Introduction:

 “Documented symptoms for long-COVID include breathlessness, fatigue, myalgia, chest pains and insomnia [5].”

 “In a study of patients with acute respiratory distress syndrome about a third of those who were previously employed were still unemployed 5-years later [7], suggesting long term disability. A dysfunctional and uncontrolled immune response can cause multi-organ damage, particularly the liver and kidneys, and disrupt the coagulation control mechanisms of the blood [8]. This can precipitate major adverse cardiovascular events which may have long-term consequences such as heart failure or hemiplegia. Data from the COVID Infection Survey study on long-COVID suggests that the risk of major adverse cardiovascular events is about ten times higher in cases with non-intensive care hospitalized patients with COVID when compared to matched controls [9]. Following treatment in critical care with acute respiratory distress syndrome, about 25% of patients have post-traumatic stress disorder (PTSD) and about 40% suffer depression [10,11].”

How this is incorporated into the model should now be clearer in the methods section:

 “Total symptom prevalence at each time point was estimated as a combination of symptomatic persons with short-term injury, which decays over time, and persons with permanent injury, which has an unchanging prevalence.”

We have also removed mention of “lung fibrosis, the sequelae of major adverse cardiovascular events like heart attacks and strokes and psychological impacts such as PTSD” in the results section as these were simply used as illustrations and not explicitly measured. In hindsight, this section could confuse readers and was therefore removed. 

Discussion

*This section is very long and less convincing.

It is difficult to make specific changes based on this feedback. We have reduced the word count of the discussion by increasing information density. We also combined the two separate sections where symptom prevalence studies were discussed to improve the flow of the discussion. 

Reviewer #2: 

*Regarding the methods section, I noticed the tense was often changed to the present tense. This is very confusing and I feel it is important to remain with the past tense at all times, as the methods are describing what you did. This is true as well for the description of your methods in the abstract. Therefore, "Both parts are combined..." should be changed to "Both parts were combined" on line 28.

The model development description in the Methods section has been put into the past tense.

*This is a very minor comment, but could be of interest. You use UTI, which I assume is Intensive Treatment Unit. As this is often referred to as the ICU in the United States, it may be of interest to simply define it the first time you use this acronym (I believe on line 99).

Done.

*In the Discussion section, on line 263 you mention, "In addition, this provides support for non-pharmaceutical interventions that reduce the transmission of the SARS-Cov-2", this is a great point. It could be of interest to provide one or several examples by finishing the sentence with, "such as..."

Done:

 “Given the socio-economic disparity in the pandemic burden, this may provide justification for spending which seeks to reduce health inequalities such as tailored public health messaging, vaccination delivery, and community access to care. In addition, this provides support for non-pharmaceutical interventions that reduce the transmission of the SARS-Cov-2, such as physical distancing and the use of face masks.”

*There appears to just be a small typo on line 267 "not return to fully to work" should be "not return fully to work"

Done.

*This is very small, but on line 287 it should be "different types of questionnaires" with an "s" at the end

Done.

*On line 279, I believe it should be "although NICE has published" instead of "have published"

Done.

*The limitations section made some very interesting points. I did feel though this section could be re-organised slightly. For example, on line 307 the discussion of recall bias is very interesting but does not fit there. I would discuss recall bias under paragraph 2 where you discuss challenges with standardized measures and recording of data.

We have made the specific change as requested.

---

## [Decision Letter · Decision Letter 1]

1 Oct 2021

PONE-D-21-16088R1A model framework for projecting the prevalence and impact of Long-COVID in the UKPLOS ONE

Dear Dr. Martin,

Thank you for submitting your manuscript to PLOS ONE. After careful consideration, we feel that it has merit but does not fully meet PLOS ONE’s publication criteria as it currently stands. Therefore, we invite you to submit a revised version of the manuscript that addresses the points raised during the review process.

Please revise.

We look forward to receiving your revised manuscript.

Kind regards,

Academic Editor

PLOS ONE

Reviewers' comments:

Reviewer's Responses to Questions

**Comments to the Author**

1. If the authors have adequately addressed your comments raised in a previous round of review and you feel that this manuscript is now acceptable for publication, you may indicate that here to bypass the “Comments to the Author” section, enter your conflict of interest statement in the “Confidential to Editor” section, and submit your "Accept" recommendation.

Reviewer #2: All comments have been addressed

Reviewer #3: All comments have been addressed

2. Is the manuscript technically sound, and do the data support the conclusions?

Reviewer #2: Yes

Reviewer #3: Yes

3. Has the statistical analysis been performed appropriately and rigorously? 

Reviewer #2: I Don't Know

Reviewer #3: Yes

4. Have the authors made all data underlying the findings in their manuscript fully available?

Reviewer #2: No

Reviewer #3: Yes

5. Is the manuscript presented in an intelligible fashion and written in standard English?

Reviewer #2: Yes

Reviewer #3: Yes

6. Review Comments to the Author

Reviewer #2: All of my original comments were addressed, thank you.

Regarding data availability. It sounds like you will make all data available but I did not see where this was uploaded or simply if you were stating that you will make all data available.

Reviewing the comments by reviewer 1, I think this reviewer made a good point that you should mention in your methods section specifically which symptoms are included under your definition of the presence of symtpoms, or at least stating "all symptoms detailed in the CIS survey, such as, ..." . I understand this must be all symptoms detailed in the survey the patients completed. It would be of interest to provide access to this survey in an annex. I should say all surveys that patients completed should be included as annexes.

Reviewer #3: Please modify the format of the abstract to standard format. That is, exclude subtitles that make things confusing.

7. PLOS authors have the option to publish the peer review history of their article (what does this mean?). If published, this will include your full peer review and any attached files.

Reviewer #2: No

Reviewer #3: No

---

## [Author Response · Author response to Decision Letter 1]

8 Nov 2021

Comment 4.

Availability of data.

The data has been freely available on the Github repository at this location:

https://github.com/Crystallize/longCOVID.

We gave the URL of this repository in two places in the submission, but it is not given in the body of the article itself as we assumed that this would appear as a link on the web page. To clarify this, I have added a sentence at the end of the methods section with the URL. If you do, in fact, provide the link separately, feel free to remove this sentence if you feel it appropriate.

6. Review Comments to the Author

Reviewer 2.

The data is available via Github from the URL given in the submission. We have added that URL to the text of the report for clarity.

Regarding the list of symptoms of long-Covid, we give details in lines 59 and 60 of the introduction. 

To clear up any confusion, the authors were in no way associated with the CIS that was the main source of data of symptom prevalence and we are therefore unable to provide copies of questionnaires etc that were used. The data that we used is only that which is publicly available from the sources given in the references.

Reviewer 3.

We have removed the headings from the abstract.

We noted that there was no guidance given on headings in the abstract and that more than half of the abstracts we examined on the Plos One site had headings, so we included them. We have no problem with them being removed if that is preferred.

---

## [Decision Letter · Decision Letter 2]

18 Nov 2021

A model framework for projecting the prevalence and impact of Long-COVID in the UK

PONE-D-21-16088R2

Dear Dr. Martin,

We’re pleased to inform you that your manuscript has been judged scientifically suitable for publication and will be formally accepted for publication once it meets all outstanding technical requirements.

Kind regards,

Academic Editor

PLOS ONE

Additional Editor Comments (optional):

Reviewers' comments:

Reviewer's Responses to Questions

**Comments to the Author**

1. If the authors have adequately addressed your comments raised in a previous round of review and you feel that this manuscript is now acceptable for publication, you may indicate that here to bypass the “Comments to the Author” section, enter your conflict of interest statement in the “Confidential to Editor” section, and submit your "Accept" recommendation.

Reviewer #2: All comments have been addressed

Reviewer #3: All comments have been addressed

2. Is the manuscript technically sound, and do the data support the conclusions?

Reviewer #2: Yes

Reviewer #3: Yes

3. Has the statistical analysis been performed appropriately and rigorously? 

Reviewer #2: I Don't Know

Reviewer #3: Yes

4. Have the authors made all data underlying the findings in their manuscript fully available?

Reviewer #2: Yes

Reviewer #3: Yes

5. Is the manuscript presented in an intelligible fashion and written in standard English?

Reviewer #2: Yes

Reviewer #3: Yes

6. Review Comments to the Author

Reviewer #2: I found the work to be well done and find it acceptable for publication.

Please not that I do have have any additional comments.

Reviewer #3: no additional comments at this point. xxxxxxxxxxxxxxxxxxxxxxxxxxxxxxxxxxxxxxxxxxxxxxxxxxxxxxxxxxxxxx

7. PLOS authors have the option to publish the peer review history of their article (what does this mean?). If published, this will include your full peer review and any attached files.

Reviewer #2: No

Reviewer #3: No

---

## [Editor Report · Acceptance letter]

23 Nov 2021

PONE-D-21-16088R2 

*A model framework for projecting the prevalence and impact of Long-COVID in the UK*

Dear Dr. Martin:

I'm pleased to inform you that your manuscript has been deemed suitable for publication in PLOS ONE. Congratulations! Your manuscript is now with our production department. 

Kind regards, 

on behalf of

Dr. Robert Jeenchen Chen 

Academic Editor

PLOS ONE